# Metabolomic Signatures Predict Seven-Year Mortality in Clinically Stable COPD Patients

**DOI:** 10.3390/ijms26136373

**Published:** 2025-07-02

**Authors:** César Jessé Enríquez-Rodríguez, Bella Agranovich, Sergi Pascual-Guàrdia, Rosa Faner, Ramon Camps-Ubach, Ady Castro-Acosta, José Luis López-Campos, Germán Peces-Barba, Luis Seijo, Oswaldo Antonio Caguana-Vélez, Diego Rodríguez-Chiaradia, Esther Barreiro, Eduard Monsó, Borja G. Cosío, Ifat Abramovich, Alvar Agustí, Carme Casadevall, Joaquim Gea

**Affiliations:** 1Hospital del Mar Research Institute, Servei de Pneumologia, Hospital del Mar, 08003 Barcelona, Spain; cesarjesse.enriquez01@alumni.upf.edu (C.J.E.-R.); spascual@hmar.cat (S.P.-G.); ramon.camps.ubach@hmar.cat (R.C.-U.); ocaguana@hmar.cat (O.A.C.-V.); darodriguez@hmar.cat (D.R.-C.); ebarreiro@researchmar.net (E.B.); carme.casadevall@upf.edu (C.C.); 2MELIS Department, Universitat Pompeu Fabra, 08003 Barcelona, Spain; 3Centro de Investigación Biomédica en Red, Área de Enfermedades Respiratorias (CIBERES), ISCiii, 28029 Madrid, Spain; rfaner@recerca.clinic.cat (R.F.); lcampos@separ.es (J.L.L.-C.); gpecesbarba@gmail.com (G.P.-B.); lseijo@unav.es (L.S.); eduardmonsomolas@gmail.com (E.M.); borja.cosio@ssib.es (B.G.C.); aagusti@clinic.cat (A.A.); 4The Ruth and Bruce Rappaport Faculty of Medicine, Technion, Israel Institute of Technology, Haifa 3525433, Israel; bellakr@technion.ac.il (B.A.); ifat.a@technion.ac.il (I.A.); 5Servei de Pneumologia (Institut Clínic de Respiratori), Hospital Clínic—Fundació Clínic per la Recerca Biomèdica, Universitat de Barcelona, 08036 Barcelona, Spain; 6Servicio de Neumología, Hospital 12 de Octubre, 28041 Madrid, Spain; ady@h12o.es; 7Unidad Médico-Quirúrgica de Enfermedades Respiratorias, Hospital Universitario Virgen del Rocío, Universidad de Sevilla, 41012 Sevilla, Spain; 8Servicio de Neumología, Fundación Jiménez Díaz, Universidad Autónoma de Madrid, 28049 Madrid, Spain; 9Servicio de Neumología, Clínica Universidad de Navarra, 28027 Madrid, Spain; 10Fundació Institut d’Investigació i Innovació Parc Taulí (I3PT), 08208 Sabadell, Spain; 11Servicio de Neumología, Hospital Son Espases, Institut d’Investigació Sanitària Illes Balears (IdISBa), Universitat de les Illes Balears, 07120 Palma, Spain

**Keywords:** COPD, mortality, metabolomics, energy, amino acids, redox, microbiota

## Abstract

Chronic Obstructive Pulmonary Disease (COPD) is a complex condition with high mortality. Early identification of patients at increased risk of death remains a major clinical challenge. This pilot study aimed to explore whether plasma metabolomic profiling could aid in the prediction of long-term (7-year) mortality and provide insight into potential underlying mechanisms. Plasma samples from 54 randomly selected stable COPD patients were analyzed using both untargeted and semi-targeted LC-MS approaches. After excluding patients with unclear death data, non-COPD-related deaths and metabolomic outliers, 41 individuals were included in the final analysis. During follow-up, 13 patients (32%) died, and 28 survived. Univariate analysis identified 12 metabolites—mainly amino acids—that differed significantly between the two groups. Functional analysis suggested a significant disruption in energy production pathways. Predictive models developed using machine learning algorithms, consisting of either ten metabolites alone or nine metabolites plus FEV_1_, achieved high accuracy for 7-year mortality prediction, with the latter model performing slightly better. Internal validation was conducted using five-fold cross-validation. While exploratory, these findings support the hypothesis that early metabolic alterations, particularly in energy pathways, may contribute to long-term mortality risk in stable COPD patients, and could complement traditional prognostic markers such as FEV_1_.

## 1. Introduction

Chronic Obstructive Pulmonary Disease (COPD) is a complex and heterogeneous condition with a highly variable clinical course [1]. While some patients remain relatively stable for years, others experience rapid deterioration and premature death, often due to respiratory failure, cardiovascular events, or cancer [2]. Early identification of patients at highest risk of mortality remains a major challenge, especially during clinically stable phases, when traditional prognostic markers offer limited predictive value [3,4].

In recent years, omics-based approaches have shown promise in improving our understanding of the biological mechanisms driving COPD progression and outcomes [5,6,7,8]. Our group previously demonstrated that specific proteomic signatures identified in stable COPD patients were associated with long-term mortality, reflecting key pathophysiological mechanisms involved in disease worsening [6]. However, proteomic data capture only part of the biological landscape. Metabolomics —integrating genetic, environmental, nutritional, microbiome-related, and lifestyle influences—offers a more comprehensive snapshot of the organism’s functional state [7,8]. Consequently, it has proven to be a particularly valuable tool for identifying systemic alterations that may not yet be clinically evident.

Several studies, including our own, have explored omics-based changes related to different COPD phenotypes [5,7,8,9], suggesting the presence of a premature senescence profile in certain patients [10]. Whether baseline metabolic profiles in stable COPD can predict future mortality, however, remains an unanswered question.

In this pilot study, we aimed to evaluate the prognostic potential of plasma metabolomic signatures obtained at baseline in a well-characterized cohort of stable COPD patients, followed prospectively over seven years. Our primary objective was to identify metabolic patterns associated with long-term mortality (7-year follow-up period after blood sample collection), which could shed light on underlying disease mechanisms and support more personalized approaches to management and prevention. In addition, given that FEV_1_ offers good specificity but limited sensitivity in predicting mortality, our second objective was to assess whether combining this functional parameter with a metabolite panel could enhance predictive performance.

## 2. Results

### 2.1. Characterization of COPD Patients

Among the 54 patients initially selected for this study, survival status and cause of death remained uncertain in seven cases. In three additional cases, death was attributed to causes considered unrelated to COPD or respiratory status (two from pancreatic cancer and one from abdominal sepsis leading to septic shock). Furthermore, metabolic data from three additional patients (all of whom were alive) were excluded as statistical outliers (see the Methods Section). Consequently, a total of 41 patients were included in the final analysis (Figure 1). Of these, 32% had died within seven years of blood sample collection. The causes of death were respiratory failure (54%), lung cancer (8%), or cardiovascular events (38%).

The main general and clinical characteristics of deceased and surviving patients are presented in Table 1. Deceased patients were slightly older than survivors. Both groups consisted of older adults, with a male predominance, consistent with national COPD demographics. The cohort predominantly presented with moderate-to-severe disease, characterized by chronic airflow obstruction and reduced lung diffusing capacity for carbon monoxide (DLco). Most patients were classified as GOLD stages II–IV and GOLD group E.

### 2.2. Differential Metabolite Profile of Deceased Patients

A total of 342 metabolites detected in at least 80% of the plasma samples were included in the initial analysis. These comprised 235 metabolites of endogenous origin, 35 derived from the microbiota, 7 with a potentially mixed origin (endogenous or microbiota-derived), 59 xenobiotics, and 6 unidentified compounds. Univariate analysis identified 12 differentially abundant metabolites (DAMs) between deceased and surviving COPD patients (Figure 2, Table 2 and Appendix A). Of these DAMs, eight were increased and four were decreased in deceased patients. They included nine endogenous metabolites, two presumably derived from the respiratory or gut microbiota, and one with a putative mixed (endogenous/microbial) origin. A post hoc analysis excluding patients who died within the first two years of follow-up (n = 4) showed that these key metabolites remained consistent in direction and similar in magnitude of change, with most retaining *p*-values < 0.1 despite the reduced statistical power (Appendix A). These results support the robustness of the original findings and suggest that the observed metabolic alterations are not solely driven by early mortality cases.

### 2.3. Biological Function Analysis

Considering significant DAMs and trend-level metabolites between deceased and surviving patients (Table 2, above and below the dashed line, respectively), nine were associated with mitochondrial function and energy metabolism (butyryl carnitine, monomethylglutarate, citrate, pyruvic acid, phosphoglyceric acid, aspartate, and creatine) either directly or indirectly (e.g., 3-methyl-2-oxovaleric acid and methylnicotinamide). Three metabolites were involved in redox homeostasis and inflammation (methylnicotinamide, L-DOPA, and L-cysteine), while two were linked to extracellular matrix remodeling (proline and hydroxyproline). In addition, two endogenous metabolites (hydroxyglutarate and 3-methyl-2-oxovaleric acid) were also related to amino acid and nitrogen metabolism. Furthermore, five plasma metabolites were found to be exclusively or predominantly of microbial origin, reflecting microbiota–host interactions and suggesting an underlying gut microbiota dysbiosis (Figure 3).

### 2.4. AI Modeling

Restricting the analysis to the 271 metabolites with complete data across all patients, the predictive model based on the top 10 ranked DAMs (Figure 4) achieved a sensitivity of 0.92 and a specificity of 0.91 when its predictions were compared with actual data (Table 3).

In contrast, when allowing the AI algorithm to freely select the best-fitting metabolites (Figure 5)—regardless of whether they were significantly different between survivors and deceased patients—model performance declined, notably in terms of sensitivity (Table 3, Appendix A).

### 2.5. Integration of FEV_1_ into the Predictive Model

When FEV_1_ was incorporated into the metabolite-based prediction model, performance improved substantially with both approaches used in this study. Using the top nine DAMs together with FEV_1_ (Figure 6), the model achieved a sensitivity of 0.83 and a specificity of 1.00. (Table 4).

In contrast, when the AI algorithm was allowed to freely select the most predictive features—including FEV_1_—(Figure 7) the resulting model reached a sensitivity of 1.00 and a specificity of 0.92 (Table 4, Appendix A).

Finally, using FEV_1_ alone with a conventional cutoff of <50% predicted [13] yielded a markedly low sensitivity (38%), despite a specificity of 86%, underscoring the limited predictive power of this traditional approach when used in isolation.

## 3. Discussion

The most relevant finding of this study is the identification of a distinct metabolomic profile in patients with stable COPD who exhibited poor survival over a 7-year follow-up period. This profile is characterized by alterations in energy and amino acid metabolism, redox imbalance, and the involvement of metabolites likely derived from the microbiota. Furthermore, we were able to derive a metabolomic signature that reasonably predicts mortality from cardiorespiratory causes within this timeframe. Notably, the predictive performance of this model improved markedly when FEV_1_ was incorporated, reaching excellent predictive accuracy.

### 3.1. Dysregulation of Energy and Mitochondrial Metabolism

Evidence of disrupted energy and mitochondrial metabolism in deceased COPD patients is supported by the observed increases in several endogenous metabolites, including butyryl-L-carnitine, monomethylglutarate, creatine, and pyruvic acid, along with strong upward trends for 3-methyl-2-oxovaleric acid, 2-phosphoglyceric acid, and aspartate. Additionally, significant decreases in 1-methylnicotinamide and citric acid may reflect complementary metabolic disturbances. Collectively, these alterations suggest dysfunction in glycolysis, fatty acid β-oxidation, and the tricarboxylic acid (TCA) cycle. These findings point to mitochondrial dysfunction and impaired energy-generating pathways as key features distinguishing stable COPD patients with poor long-term survival. In the sections that follow, each of these metabolites is discussed individually.

*Acylcarnitines*, *such as butyryl-L-carnitine*, play a crucial role in shuttling fatty acids into the mitochondria for β-oxidation. Elevated levels of these molecules may reflect inefficient or impaired function of this energy pathway. Alternatively, they may indicate disturbances in branched-chain amino acid (BCAA) metabolism (e.g., isoleucine, valine, or threonine), or reflect inflammatory activation [14]. Moreover, acylcarnitines have also been described as products of microbial metabolism. Previous work from our group demonstrated that changes in circulating acylcarnitine profiles can differentiate COPD patients from smokers without airflow obstruction [15], likely reflecting broader disruptions in lipid metabolism. Other studies have similarly reported increased levels of specific acylcarnitines in both lung tissue and serum of patients with COPD compared to healthy controls [16,17,18]. Notably, hypercapnia—a frequent condition in advanced COPD—has been shown to upregulate acylcarnitine production in blood monocytes and macrophages [19].

*Monomethylglutarate* is a methylated derivative of glutaric acid involved in both lipid metabolism (notably β-oxidation) and amino acid catabolism—particularly of lysine, hydroxylysine, tryptophan, and leucine—thereby exerting a direct impact on energy metabolism. Its elevated levels may reflect mitochondrial dysfunction and concomitant oxidative stress. To our knowledge, this is the first report of altered monomethylglutarate levels in COPD patients.

*Creatine* is a nitrogenous compound synthesized from amino acids. It circulates in both free form and as phosphocreatine, a rapidly mobilizable reservoir of high-energy phosphates within metabolically active tissues such as skeletal muscle. Notably, skeletal muscle dysfunction and loss of muscle mass are among the most prominent systemic manifestations of COPD [20,21], and both are well-established predictors of poor survival [22]. The elevated plasma creatine levels observed in our study may reflect chronic leakage from atrophic muscles, possibly as a consequence of ongoing muscle atrophy or catabolic processes in patients who will die over the subsequent seven years. In contrast to our findings, reduced plasma creatine levels have been associated with 1-year mortality in COPD [23], possibly indicating a more advanced stage of muscle wasting and a shorter survival horizon in that population.

*Pyruvic acid* is a key intermediate in cellular metabolism, serving as the end product of glycolysis. Under aerobic conditions, it is transported into mitochondria and converted to acetyl-CoA, entering the tricarboxylic acid (TCA) cycle to support oxidative phosphorylation. Additionally, pyruvate participates in gluconeogenesis and serves as a precursor for the synthesis of alanine. Previous studies have reported altered pyruvate levels in various biological samples from COPD patients, including plasma [24], urine [25], and peripheral blood mononuclear cells [26], reinforcing the hypothesis of significant alterations in energy metabolism.

*3-Methyl-2-oxovaleric acid* is a catabolic product of branched-chain amino acids (BCAAs), whose derivatives can enter the TCA cycle and contribute to ATP production. Elevated levels of this metabolite may reflect reduced utilization of TCA intermediates, suggestive of impaired mitochondrial energy metabolism and/or inefficient amino acid oxidation. Although this metabolite has not previously been associated with COPD, it has been linked to metabolic and cardiovascular disorders [27], which are known to impact COPD progression and prognosis.

*2-Phosphoglyceric acid (2-PG)*, in turn, is a key intermediate in the glycolytic pathway. Its accumulation has been associated with cellular stress responses, where it may serve as a compensatory mechanism in response to impaired energy production via aerobic metabolism. However, to our knowledge, alterations in plasma 2-PG levels have not yet been reported in patients with COPD.

*Aspartate*, a proteinogenic amino acid, also plays a role in the urea cycle as a nitrogen donor and serves as a precursor for nucleotide synthesis. It can be transaminated into oxaloacetate, a critical intermediate of the TCA cycle, thereby contributing to energy metabolism. Moreover, elevated aspartate levels may also reflect immune system activation. In COPD, altered aspartate concentrations have previously been associated with the degree of airflow obstruction and the presence of skeletal muscle dysfunction [28]. Increased aspartate levels have also been reported in serum and urine from patients with severe emphysema, advanced COPD, or nutritional deficiencies, likely indicating increased protein catabolism [8,29].

*1-Methylnicotinamide (MNA)* is a metabolite of vitamin B3 involved in multiple biological processes, including energy metabolism, redox balance, and modulation of inflammatory responses [30]. It has also been implicated in cellular senescence and the pathophysiology of chronic diseases. Nicotinamide metabolism is known to be altered in COPD and has previously been associated with disease progression [25]. Notably, the therapeutic potential of nicotinamide derivatives—such as nicotinamide riboside—has gained attention, as recent studies have shown that their administration may attenuate pulmonary inflammation in COPD patients [31].

A parallel interpretation applies to the decreased levels of citrate observed in patients who subsequently died. Citrate is a key intermediate in the early stages of the TCA cycle and plays a central role in energy production. Beyond its mitochondrial function, it also contributes to the biosynthesis of carbohydrates, fatty acids, cholesterol, NAD, NADH, and certain amino acids such as aspartate. The concurrent alterations in both citrate and aspartate levels may reflect mitochondrial dysfunction, possibly as a downstream consequence of sustained systemic inflammation.

Together, these metabolic alterations point toward a central role for mitochondrial dysfunction and impaired energy metabolism in determining long-term mortality risk among patients with stable COPD. This bioenergetic disruption may reflect the cumulative effects of chronic inflammation, muscle wasting, and systemic metabolic stress.

### 3.2. Alterations in Amino Acid and Nitrogen Metabolism

In addition to the disruption of energy-generating pathways, we also identified changes suggestive of altered amino acid and nitrogen metabolism. These were reflected by increased levels of N-phenylacetylglutamine (PAGIn) and hydroxyproline, along with similar trends for proline, 3-methyl-2-oxovaleric acid, and aspartate. Conversely, reduced levels of L-cystine and a downward trend in 2-hydroxyglutarate may also signal dysfunction in these same metabolic routes.

*N-Phenylacetylglutamine* is a conjugate of phenylacetic acid and L-glutamine, and its levels tend to rise when the urea cycle is overwhelmed, functioning as part of an alternative nitrogen elimination pathway. It may also originate from intestinal microbiota-derived precursors that are subsequently metabolized in the liver. Elevated concentrations of this metabolite have been linked to systemic inflammation, increased platelet aggregation, and thrombus formation, potentially contributing to cardiovascular complications. Notably, a recent study also associated higher N-phenylacetylglutamine levels with an increased risk of developing COPD [32].

*Hydroxyproline and proline* are key structural components of collagen and serve as markers of the turnover of this structural molecule. Elevated levels of these metabolites may reflect active remodeling of the extracellular matrix [33], with a possible predominance of matrix degradation, apoptosis, and autophagy-related cellular breakdown [34]. Altered proline metabolism has been reported in various tissues of COPD patients. For instance, members of our group previously demonstrated increased hydroxyproline content in lung tissue from patients with emphysema compared to those without this structural abnormality [35]. Additionally, elevated concentrations of the peptide acetyl–proline–glycine–proline (AcPGP) in sputum have been shown to correlate with emphysema severity and airflow limitation in COPD [36].

*2-Hydroxyglutarate* is a secondary metabolite primarily derived from the catabolism of certain amino acids, such as glutamate. At low concentrations, it may contribute to the epigenetic regulation of enzyme activity or reflect oxidative stress. A recent serum metabolomic study identified L-2-hydroxyglutarate as a potential biomarker for predicting acute exacerbations in COPD, suggesting a potential role in the biological characterization of the frequent exacerbator phenotype [37].

These findings highlight the presence of significant disturbances in amino acid and nitrogen metabolism among COPD patients with poor long-term outcomes. Such alterations may reflect increased protein catabolism, impaired nitrogen disposal, and disrupted extracellular matrix turnover—mechanisms that likely contribute to both systemic deterioration and disease progression.

### 3.3. Redox Imbalance and Neuroendocrine Dysregulation

The observed decrease in *L-cystine*—a non-essential amino acid involved in the transsulfuration pathway and a precursor in the biosynthesis of glutathione—likely reflects heightened oxidative stress in patients who died during the 7-year follow-up period. To date, no studies have specifically reported plasma alterations of L-cystine in patients with COPD.

Similar pathophysiological implications may be inferred from the alterations observed in *1-methylnicotinamide*, monomethylglutarate, and threitol. Oxidative stress is widely recognized as a central mechanism in COPD, contributing to both pulmonary damage and systemic manifestations [1].

Differences in *L-DOPA* levels were also observed between patients who died and those who survived. These alterations may reflect oxidative stress, particularly within the neuroendocrine system, as L-DOPA is a non-proteinogenic amino acid that serves as a precursor to catecholamines and functions as an indirect neurotransmitter. Elevated levels of L-DOPA could be linked to dysregulation of dopaminergic signaling, with potential effects on vascular and bronchial tone. Alternatively, its increase may have been partially driven by β-agonist therapy, a possibility supported by animal studies [38].

Taken together, the alterations observed in L-cystine, 1-methylnicotinamide, monomethylglutarate, threitol, and L-DOPA provide further evidence of systemic redox imbalance and oxidative stress in COPD patients with poor long-term survival. These changes likely reflect a convergence of mitochondrial dysfunction, impaired antioxidant defenses, and potential neuroendocrine dysregulation. Oxidative stress is a well-established contributor to COPD pathogenesis and progression, and our findings support its role not only in local pulmonary injury but also in systemic processes that may influence overall prognosis.

### 3.4. Microbiota-Derived Metabolites and Host–Microbiota Interactions

Our findings also suggest that host–microbiota interactions may play a role in the poor long-term prognosis of certain COPD patients. This is reflected by altered levels of several metabolites likely derived from bacterial fermentation—primarily in the gut, though potentially also in the respiratory tract. Notably, differences were observed in indoxyl sulfate, N-phenylacetylglutamine, and 4-aminobenzoate, alongside upward trends in phenol and threitol. These metabolites are commonly linked to microbial activity and may indicate dysbiosis with increased systemic absorption of potentially harmful bacterial products.

*Indoxyl sulfate* is a uremic toxin produced from dietary tryptophan by intestinal bacteria, followed by hepatic sulfation. Its accumulation in the bloodstream has been associated with endothelial dysfunction and cardiovascular risk [39]. 4-Aminobenzoate in turn, serves as a microbial precursor in folic acid synthesis, a key cofactor in the production of hemoglobin and structural proteins.

*Phenol*, a byproduct of microbial degradation of aromatic amino acids such as tyrosine or phenylalanine, and threitol, a carbohydrate fermentation product, both showed a tendency to increase in patients with poor outcomes. These patterns are also consistent with a shift toward fermentative bacterial species, possibly contributing to systemic toxicity and inflammation.

### 3.5. Metabolic Signature of Poor Prognosis

The metabolomic signature derived in this study demonstrated excellent predictive performance for 7-year mortality, achieving both sensitivity and specificity values above 90% when the model was restricted to DAMs.

While FEV_1_ is a well-established clinical predictor of mortality, our findings confirm that, when used in isolation, it offers high specificity but limited sensitivity—typically in the range of 0.35–0.40 [13,40]. However, when FEV_1_ is combined with nine selected metabolites—maintaining a comparable number of variables—the predictive capacity improved substantially. In particular, the AI-driven free selection model incorporating FEV_1_ achieved near-perfect sensitivity while preserving high specificity.

### 3.6. Strengths and Potential Limitations

One of the main strengths of this study is the rigorous collection, handling, and storage of biological samples, all obtained from reference hospitals and associated research centers within the Spanish Network of Excellence for Research in Respiratory Diseases (CIBERES). This high standard of biobanking ensures the quality, validity, and reproducibility of omics data. In parallel, these centers also provided robust clinical phenotyping, including accurate documentation of survival status and cause of death for each patient.

Another notable strength lies in the integration of metabolomic data with a well-established functional parameter, FEV_1_. This multimodal approach substantially enhanced the predictive accuracy for long-term mortality in stable COPD patients. Moreover, the exceptionally long follow-up period—seven years—is a rare feature in studies involving omics biomarkers, adding substantial value to the prognostic analysis.

An additional strength is the biological coherence of the findings. The differentially abundant metabolites were not treated as isolated markers but rather grouped into interconnected functional pathways (e.g., mitochondrial metabolism, amino acid catabolism, redox homeostasis, extracellular matrix remodeling, and host–microbiota interactions), enhancing their mechanistic plausibility.

A potential limitation of this study is the relatively small sample size, which is inherent to its pilot nature. However, this is counterbalanced by a carefully designed patient selection process, high-quality data, long-term follow-up, and the use of robust statistical and machine learning methods that minimize overfitting. Importantly, the risk of selection bias was reduced by enrolling participants at the outset of this study without stratification by demographic or clinical variables and independently of outcome status.

Another limitation is the absence of multiple testing correction in the univariate comparison of clinical groups prior to modeling. Nonetheless, in exploratory metabolomics studies with moderate sample sizes and low individual effect sizes, overly stringent correction methods (e.g., FDR < 0.05) may increase the risk of discarding biologically meaningful signals. Instead, we opted for an integrative strategy that combined nominal significance (uncorrected *p* ≤ 0.05) with biological relevance, prior evidence, and metabolic interconnectivity. This approach supports a pathway-based interpretation, strengthens the hypothesis-generating potential of our findings, and provides a sound basis for future validation studies in independent cohorts or through targeted metabolomics approaches.

Finally, although the cohort consisted exclusively of clinically stable COPD patients, we acknowledge that post-baseline clinical events—such as exacerbations, comorbidities, or treatment changes—could have influenced long-term outcomes. However, since all plasma samples were collected at baseline, these subsequent events could not have affected the initial metabolomic profiles and therefore cannot confound the associations a priori. To address the possibility of bias from early deaths, we conducted a post hoc sensitivity analysis excluding patients who died within the first two years (n = 4). The resulting subcohort (n = 38) showed consistent direction and magnitude of metabolite differences, reinforcing the robustness of the original findings (see Appendix A).

We also acknowledge the limitation of not having uniformly available longitudinal treatment data—particularly regarding inhaled corticosteroid (ICS) use—which precluded treatment-based stratification. This reflects the multicenter, observational nature of the cohort and underscores the need for prospective studies with repeated biosampling and structured follow-up. To this end, we are currently planning a follow-up investigation in a larger, independent cohort of clinically stable COPD patients. This study will apply a targeted metabolomics approach to quantify the most promising biomarkers identified in the present work, with the aim of confirming their reproducibility, prognostic value, and potential clinical utility under standardized conditions and across diverse clinical settings

## 4. Materials and Methods

### 4.1. Participants and Ethics

This pilot prospective study initially included 54 clinically stable Caucasian patients with COPD, randomly selected—without any prior stratification by clinical, demographic, or survival-related variables—from the group of non-frequent exacerbators within the BIOMEPOC cohort at study inception. Patients were recruited between 2015 and 2018 from seven tertiary university hospitals across Spain and have been clinically followed since enrollment. A detailed description of the cohort’s clinical procedures has been published previously [41].

This study was approved by the institutional Ethics Committee (ref. 2014/5895/l), and all procedures adhered to the Declaration of Helsinki as well as international and local regulations. Written informed consent was obtained from all participants prior to any clinical evaluation or sample collection. Some patients had also participated in other studies investigating blood biomarkers for unrelated objectives.

COPD was diagnosed based on a history of tobacco use and/or exposure to noxious inhalants, combined with persistent airflow limitation, defined as a post-bronchodilator FEV_1_/FVC ratio < 0.7 [1]. Clinical stability in turn, was defined as the absence of exacerbations for at least three months prior to enrollment. Exclusion criteria included frequent exacerbations (≥2 moderate-to-severe acute episodes in the previous year) [1], presence of other chronic pulmonary or inflammatory diseases, and deaths due to the COVID-19 pandemic. Data on the 7-year mortality and its causes were collected from participating centers, and a survival stratification variable was generated based on follow-up. Participants who were alive or had survived ≥2555 days were classified as “alive”, whereas those who died before 7-year mark were considered “deceased”. This study complies with the STROBE guidelines for reporting observational research.

### 4.2. Blood Sample Collection

Peripheral blood samples were collected by venipuncture after an overnight fast into K_3_-EDTA-coated tubes. Samples were centrifuged at 1500× *g* for 15 min at 4 °C to separate plasma, which was subsequently aliquoted and stored at −80 °C until metabolomic analysis.

### 4.3. Metabolomics Workflow

The complete metabolite extraction protocol used in our laboratory has been previously described [15].

#### 4.3.1. Metabolite Collection and LC-MS Parameters

Briefly, a pooled quality control (QC) sample was prepared by combining 10 µL from each individual plasma sample. Then, 10 µL of each plasma sample were mixed with 190 µL of cold extraction buffer (Cambridge Isotope Laboratories, Tewksbury, MA, USA). Samples were vortexed for 10 min at 4 °C and centrifuged at maximum speed for 15 min. An 80 µL aliquot of the resulting supernatant was transferred into HPLC vials and stored at −80 °C until LC-MS analysis. QC samples were injected every 10 runs to monitor intra-batch variability.

Chromatographic separation was performed using a Thermo Vanquish Flex UPLC system coupled to an Orbitrap Exploris 240 mass spectrometer (Thermo Fisher Scientific, Waltham, MA, USA). Metabolites were separated on a ZIC-pHILIC column (SeQuant, Merck, NJ, USA), with 5 µL of plasma extract injected per run under a 15-min mobile phase gradient. The flow rate was set at 0.2 mL/min, and the column was maintained at 45 °C. Data acquisition was performed using Thermo Xcalibur 4.4.

#### 4.3.2. Metabolite Identification and Quantification

These were carried out using complementary semi-targeted and untargeted approaches. In the semi-targeted workflow, metabolite peak areas were determined using Thermo TraceFinder^TM^ 5.1, and identification was based on our in-house mass spectral (MS) library. For untargeted analysis, Compound Discoverer 3.3 (Thermo Fisher Scientific) was used. Retention time alignment across datasets was achieved using the ChromeAlign node, with the pooled QC sample serving as reference for quality control and batch normalization.

Compound identification followed a hierarchical strategy: (1) matching mass and retention time with the in-house library, (2) comparing fragmentation spectra to the mzCloud database, and (3) for non-fragmented features, querying HMDB, KEGG, and BioCyc using Compound Discoverer’s mzlogic score >50, while limiting potential candidate matches to fewer than three [15]. Only those metabolites detected in ≥80% of plasma samples were retained for the exploratory phase of the analysis.

#### 4.3.3. Metabolomic Outlier Identification and Quality Control

The LC-MS output was assessed using a proprietary Python (version 3.10) script for quantitative outlier detection and quality control. Rather than relying exclusively on visual exploration via principal component analysis (PCA), Euclidean distance calculations were used to quantitatively evaluate data quality and integrity. Data processing and formatting were performed using Pandas (version 2.0.3), and a log transformation was applied with NumPy (version 1.25.2) to reduce distributional skewness. The dataset was then standardized using Scikit-learn’s StandardScaler (version 1.3.0), and pairwise Euclidean distances were computed. To visualize these distances and facilitate outlier detection, a heatmap and hierarchical clustering dendrogram were generated using Seaborn (version 0.12.2) and Plotly (version 5.15.0). Based on this analysis, three outlier samples were identified and excluded from further analysis.

### 4.4. Data Analysis

#### 4.4.1. Sample Size Calculation

The minimum required sample size for this study was estimated based on a previous proteomics-based mortality study published by our group [6], using the GRANMO software package (version 8.0, FIMIM, Barcelona, Spain). A 10% loss to follow-up was assumed in the calculation.

#### 4.4.2. Clinical Data Analysis

Continuous variables were summarized as either median (interquartile range) or mean ± standard deviation, depending on data distribution. Categorical variables were reported as absolute and relative frequencies. The Shapiro–Wilk test was used to assess the normality of continuous variables, conducted using R (version 4.5.0). Between-group comparisons were conducted using t-tests for normally distributed variables, or the Mann–Whitney U test otherwise. Categorical variables were compared using chi-squared or Fisher’s exact tests, as appropriate. All analyses were performed using SPSS v25.0 (IBM, Chicago, IL, USA) and the CreateTableOne function from the R package tableone (version 0.13.2).

#### 4.4.3. Metabolomic Data Preprocessing and Filtering

Raw LC/MS data files were processed using Compound Discoverer version 3.3 to calculate total ion intensity per sample. Metabolite intensities were normalized to the total signal of each sample. Following missing value handling, a log_2_ transformation was applied using a pseudocount (1 × 10^−9^) to stabilize variance. Two strategies were employed to address missing values: (1) complete-case analysis, retaining only metabolites without any missing values across samples, and (2) imputation, where missing values were replaced with 95% of the lowest observed value.

To focus on potentially relevant endogenous and/or microbiota-associated biomarkers, xenobiotics (chemical compounds of non-human and non-microbial origin) were excluded based on classifications from the HMDB (https://hmdb.ca, accessed on 14 April 2025) and PubChem (https://pubchem.ncbi.nlm.nih.gov, accessed on 14 April 2025). Metabolites considered unclassifiable were retained.

#### 4.4.4. Covariate Adjustment and Differential Analysis

To adjust for latent sources of variability, surrogate variable analysis (SVA) was performed using the sva package in R, considering the 10% most variable metabolites based on their interquartile range (IQR). The analysis included a full model with 7-year survival status, age, and post-bronchodilator FEV_1_ as covariates, and a reduced model excluding survival status. The surrogate variables (SVs) estimated by SVA were incorporated as additional covariates into the subsequent linear modeling.

Differential abundance analysis was then conducted using the limma package, which fits linear models for each metabolite. Moderated t-tests with empirical Bayes variance shrinkage were applied. The models were adjusted for age, post-bronchodilator FEV_1_, and the SVA-derived surrogate variables to enhance the robustness of the findings by reducing the influence of potential unmeasured confounders.

At this stage, no multiple testing correction was applied, as the primary objective was feature selection for downstream predictive modeling. A post hoc analysis was also performed after excluding patients who died within the first two years of follow-up (n = 4), following the same approach, to assess the robustness of the findings beyond early mortality.

#### 4.4.5. Data Export for Machine Learning

Several data matrices were generated for machine learning analyses. These included only metabolites with *p* values < 0.05 in the differential analysis, using complete-case and imputed datasets, with and without adjustment for FEV_1_ as a covariate.

### 4.5. Generation of Predictive Models

#### 4.5.1. Model Development

Predictive models for 7-year mortality were developed using Random Forest (RF) with a conformal prediction variant, which enables not only robust classification but also a quantifiable estimation of prediction reliability. The analysis was conducted using Flame (version 1.2.2), a specialized, open-access software tool developed at our center for predictive modeling (https://phi.upf.edu/phi/index.php/software/flame/, accessed on 7 May 2025) [42].

#### 4.5.2. Initial Model Fitting

Data were processed using standard scaling. To address class imbalance, the Instance Hardness Threshold method was applied as a combined under/oversampling strategy. A conformal RF model was implemented with a confidence level of 0.8, and K-best feature selection was performed using the “auto” criterion to determine the optimal number of features. The RF algorithm used 200 trees, with class weights set to “balanced” in accordance with the preprocessing strategy. The maximum number of features for each tree split was set to the square root of the selected feature count, and a minimum of two samples per node was required for splits. Out-of-the-bag samples were used internally to estimate generalization performance. A random seed of 46 was applied to ensure reproducibility. Conformal prediction was performed using Aggregated Conformal Predictors (ACPs). Normalization of ACPs was achieved through a K-nearest neighbors (KNN) method using 15 nearest neighbors. Each ACP was derived from 10 individual models, with bootstrap sampling applied to select calibration sets. Final *p*-values were aggregated using the median.

#### 4.5.3. Internal Validation

The models were internally validated using 5-fold cross-validation. To assess the contribution of individual metabolites to model performance, a permutation importance technique was employed [42].

#### 4.5.4. Model Evaluation

Model predictions were compared against actual survival outcomes using a confusion matrix, from which the following performance metrics were calculated: sensitivity (SE), specificity (SP), positive predictive value (PPV), negative predictive value (NPV), overall accuracy (Acc), and the Matthews correlation coefficient (MCC, also referred to as the phi coefficient, φ or r_φ_). As in our previous pilot study [6], both the observed mortality rate in our cohort and data from a larger reference population of approximately 9000 patients from the COPD Cohorts Collaborative International Assessment (3CIA) [12] (see Appendix C) were used to compute PPV, NPV, and Acc.

#### 4.5.5. Biological Function Analysis

Identified metabolites were contextualized biologically by mapping them to metabolic pathways using the SMPDB, KEGG, and HMDB databases. Metabolite-metabolite interaction networks were visualized using Cytoscape software (v3.10.2, www.cytoscape.org, accessed on 14 May 2025).

## 5. Conclusions

Taken together, our results indicate that patients with stable COPD who will die in the following years are characterized by more pronounced dysregulation of energy production pathways, redox homeostasis, and amino acid and nitrogen metabolism, as well as by specific interactions with their own microbiota, when compared to those who remain alive beyond this period. This suggests that the simultaneous presence of mitochondrial dysfunction, active pulmonary tissue remodeling, chronic oxidative stress/inflammation, and dysbiosis constitutes an early metabolic footprint associated with worse outcomes, even in clinically stable individuals. Moreover, we identified a metabolomic signature with high predictive value for mortality over the subsequent years—particularly when combined with FEV_1_.

## Figures and Tables

**Figure 1 ijms-26-06373-f001:**
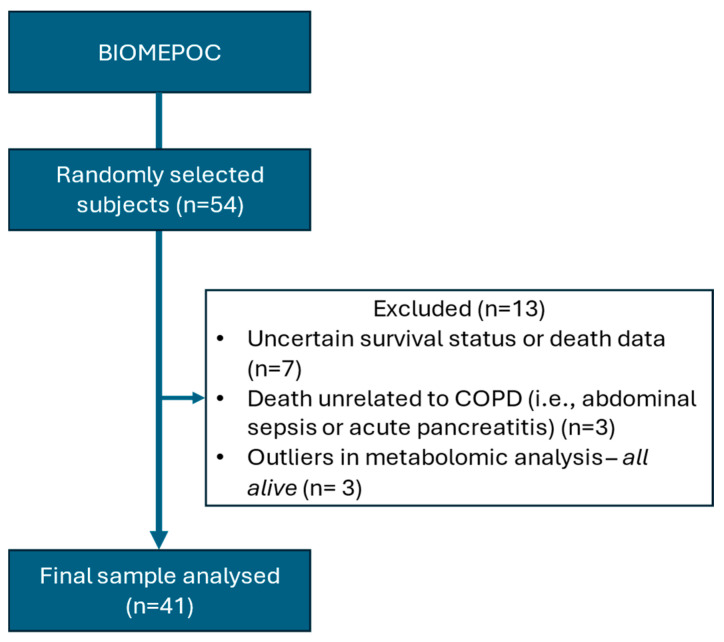
Study flowchart.

**Figure 2 ijms-26-06373-f002:**
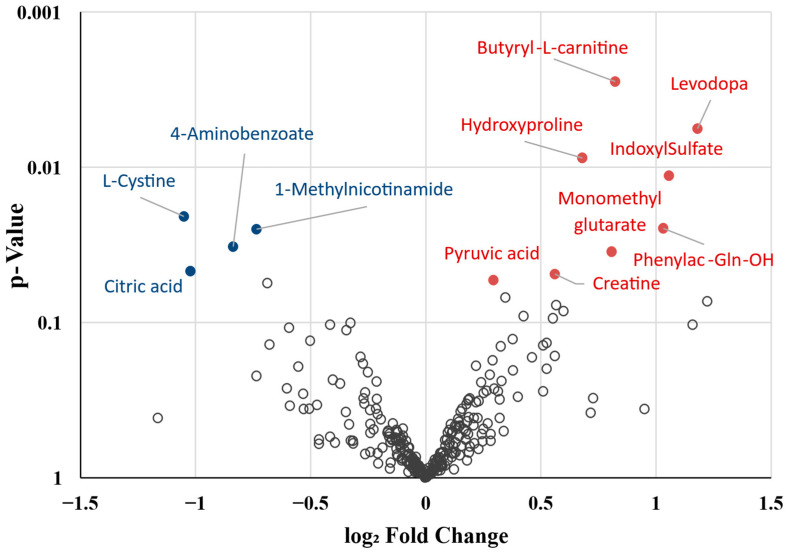
Volcano plot showing over- and underrepresented metabolites in deceased COPD patients compared to survivors. Red circles indicate significantly overrepresented metabolites, while blue circles represent significantly underrepresented metabolites.

**Figure 3 ijms-26-06373-f003:**
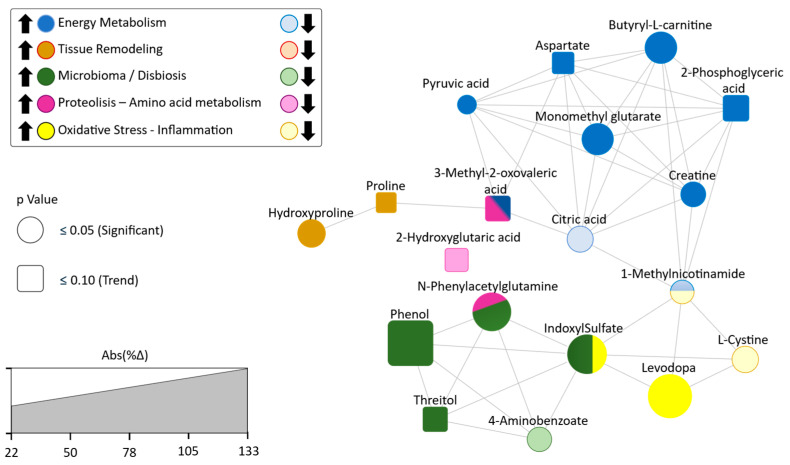
Network representation of differentially abundant metabolites (DAMs), either significant (circles) or trending (squares), between deceased and surviving COPD patients after a 7-year follow-up. Node color indicates the main biological function. The intensity of the node color reflects the direction of change in deceased patients (darker tones indicate higher levels; lighter tones, lower levels). Node size represents the magnitude of the difference (expressed as the absolute percentage difference in abundance between groups). Edges indicate that two metabolites are involved in one or more shared physiological or pathophysiological processes, based on curated annotations from SMPDB, KEGG, and HMDB.

**Figure 4 ijms-26-06373-f004:**
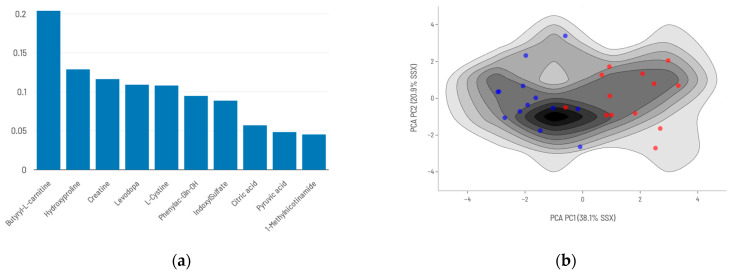
Conventional analysis model: (**a**) Top 10 differentially abundant metabolites (DAMs) and their relative importance to mortality prediction. (**b**) Density maps of survival (blue)/death (red) prediction models’ training series using the top 10 DAMs from conventional analysis.

**Figure 5 ijms-26-06373-f005:**
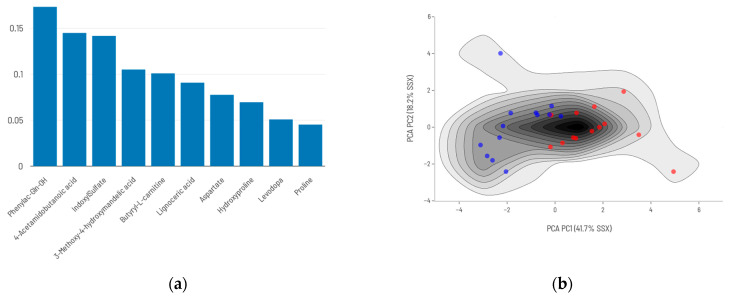
AI-selected metabolites model: (**a**) AI-selected set of 10 metabolites and their relative importance in the predictive mode. (**b**) Density maps of survival (blue)/death (red) prediction models’ training series using AI-free choice of metabolites.

**Figure 6 ijms-26-06373-f006:**
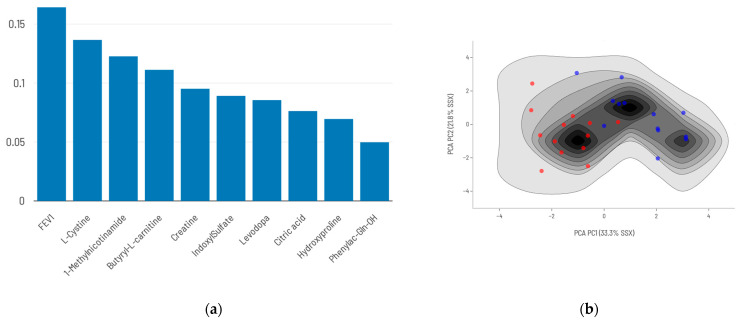
Conventional analysis plus FEV_1_ model: (**a**) Top 9 DAMs combined with FEV_1_ and their relative contribution to mortality prediction. (**b**) Density maps of survival (blue)/death (red) prediction models’ training series using the top 9 DAMs plus FEV_1_.

**Figure 7 ijms-26-06373-f007:**
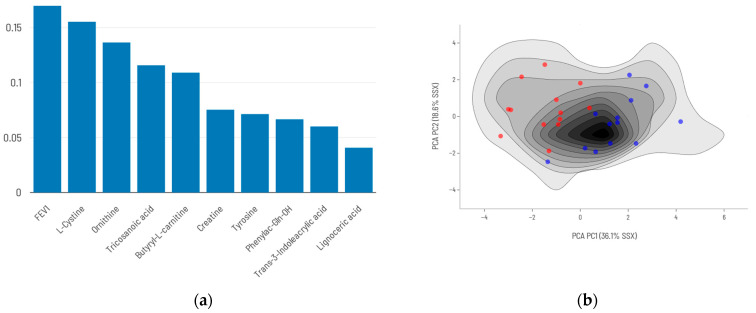
AI-selected metabolites plus FEV_1_ model: (**a**) AI-selected set of 9 metabolites plus FEV_1_ and their relative importance in the predictive model. (**b**) Density maps of survival (blue)/death (red) prediction models’ training series using AI-selected metabolites plus FEV_1_.

**Table 1 ijms-26-06373-t001:** Clinical data of deceased and surviving COPD patients at blood sample collection.

	Surviving (n = 28, 68%)	Deceased (n = 13, 32%)	*p*-Value
Age, yrs.	66.2 ± 9.0	73.4 ± 5.2	**0** **.** **01**
Sex			0.12
-Male, n (%)	15 (58)	11(42)	
-Female, n (%)	13 (87)	2 (13)	
Smoking status			0.92
-Active smoker, n (%)	13 (65)	7 (35)	
-Ex-smoker, n (%)	15 (71)	6 (29)	
Pack Years	56 ± 24	58 ± 29	0.83
BMI, kg/m^2^	26.3 ± 5.9	24.4 ± 4.0	0.32
Post-BD FEV_1_, % pred.	48.5 [39.0–67.0]	33.0 [23.0–37.0]	**<0.01**
Post-BD FEV_1_/FVC, % pred.	48.6 [38.5–59.0]	39.0 [34.0–56.0]	0.26
DLco, % pred	52.3 ± 18.5	38.2 ± 19.8	0.06
GOLD I–IV, n (%)			0.15
-I (10%)	4 (100)	0 (0)	
-II (27%)	9 (82)	2 (18)	
-III (41%)	11 65)	6 (35)	
-IV (22%)	4 (44)	5 (56)	
GOLD A-D, n (%)			0.21
-A (20%)	7 (88)	1 (12)	
-B (20%)	7 (88)	1 (12)	
-E (60%)	14 (56)	11 (44)	
Charlson score	3.0 [2.5–4.0]	3.0 [2.8–4.3]	0.67

Values are expressed as mean ± standard deviation, median [interquartile range], or n (%), as appropriate, for each group of COPD patients. Abbreviations: BMI: body mass index; BD: bronchodilator; FEV_1_: forced expiratory volume in 1 s; FVC: forced vital capacity; DLco: diffusing capacity for CO; GOLD: global initiative for Chronic Obstructive Pulmonary Disease.

**Table 2 ijms-26-06373-t002:** Panel of the 12 metabolites with significantly different abundance between deceased and surviving patients (above the dashed line), and those showing a strong trend toward differential abundance (below the dashed line).

Metabolite	log_2_FC	%Δ	*p*-Value	Predominant Source	Taxonomy	Function/Action
↑ Butyryl-L-carnitine	0.8233	76.95	<0.01	Endogen	Fatty acid ester (short chain acylcarnitine)	Lipid β-oxidation, energy production, amino acid catabolism
↑ L-DOPA	1.1807	126.69	0.01	Endogen	Amino acids, peptides and analogues	Catecholamine intermediate, amino acid metabolism
↑ Hydroxyproline	0.6793	60.14	0.01	Endogen	Amino acids, peptides and analogues	Peptide structure of collagen, tissue remodeling, amino acid metabolism
↑ Indoxyl sulfate	1.0562	107.94	0.01	Microbiota	Sulfated aromatic compound	Cardiotoxin and uremic toxin, Endothelial damage
↓ L-Cystine	−1.0506	−51.72	0.02	Endogen	Amino acids, peptides and analogues	Amino acid metabolism, redox homeostasis
↑ N-Phenylacetylglutamine (PAGIn)	1.0313	104.38	0.02	Mixed Source	Amino acids, peptides and analogues	Amino acid metabolism, Nitrogenous excretion, ↑ Platelet aggregation and thrombosis
↑ Methylnicotinamide (MNA)	−0.7350	−39.92	0.03	Endogen	Nicotinamide derivative	Nicotinate and nicotinamide metabolism, energy production, redox balance
↓ 4-Aminobenzoate	−0.8362	−43.99	0.03	Microbiota	Benzoic acids and derivatives	Folic acid synthesis
↑ Monomethylglutarate	0.8077	75.05	0.03	Endogen	Fatty Acid ester	Lipid β-oxidation, energy production, amino acid metabolism, redox balance
↓ Citric acid	−1.0226	−50.78	0.05	Endogen	Tricarboxylic acids and derivatives	TCA cycle, energy production, amino acid metabolism
↑ Creatine	0.5618	47.61	0.05	Endogen	Amino acids, peptides and analogues	Energy reserve, amino acid metabolism
↑ Pyruvic acid	0.2938	22.58	0.05	Endogen	α-Keto acids and derivatives	Glycolysis end-product, TCA cycle, urea cycle, energy production, amino acid metabolism
≈↓ 2-Hydroxyglutarate	−0.6872	−37.89	0.06	Endogen	Short-chain hydroxy acids and derivatives	Butanoate and amino acid metabolism, epigenetic regulator
≈↑ Proline	0.3449	27.01	0.07	Endogen	Amino acids, peptides and analogues	Peptide structure of collagen, tissue remodeling, amino acid metabolism
≈↑ Phenol	1.2227	133.39	0.08	Microbiota	Benzenoids, phenolic compound	Sulfate/sulfite metabolism, dysbiosis
≈↑ 3-Methyl-2-Oxovaleric acid	0.5672	48.17	0.08	Endogen	Short-chain keto acids and derivatives	Amino acid metabolism, energy production
≈↑ 2-Phosphoglyceric acid (2-PG)	0.5983	51.39	0.09	Endogen	Carbohydrates and their conjugates	Glycolysis, gluconeogenesis, energy production
≈↑ Aspartate	0.4243	34.19	0.09	Endogen	Amino acids, peptides and analogues	Energy metabolism and urea cycle, amino acid metabolism, nicotinate and nicotinamide metabolism
≈↑ Threitol	0.5532	46.73	0.09	Mixed source	Carbohydrates and their conjugates	Redox balance, dysbiosis

Abbreviations and symbols: log_2_FC: log_2_ fold change; %Δ: percentage difference (based on log_2_ fold change); ↑: increased in deceased patients; ↓: decreased in deceased patients; ≈: statistical trend (0.05 < *p*-value < 0.10). Comparisons were calculated using moderated t-tests with empirical Bayes adjustment, correcting for age, FEV_1_, and latent surrogate variables.

**Table 3 ijms-26-06373-t003:** Comparison of prediction models using conventional DAMs versus AI-based free metabolite selection.

	Conventional	AI Free Choice
**Fitting**		
Sensitivity–Specificity–MCC	1.00	1.00
Coverage	0.96	1.00
**Prediction**		
Sensitivity	0.92	0.75
Specificity	0.91	0.90
MCC	0.83	0.65
Coverage	0.92	0.85
PPV (rep/our)	0.90/0.83	0.87/0.78
NPV (rep/our)	0.93/0.96	0.81/0.89
Accuracy (rep/our)	0.92/0.91	0.83/0.85

PPV, NPV, and accuracy were calculated based on either (a) the mortality observed in our COPD cohort (32%) [our] or (b) the mortality rate previously reported [rep] in the Spanish COPD population (47%) [11], which aligns with mortality predictions derived from the 3CIA initiative detailed in Appendix C [6,12]. Abbreviations: MCC: Matthews correlation coefficient; PPV: positive predicted value; NPV: negative predicted value; rep: reported mortality; our: our own mortality; AI: artificial intelligence.

**Table 4 ijms-26-06373-t004:** Comparison of prediction models using conventional DAMs versus AI-based free metabolite selection combined with FEV_1_ integration.

	Conventional + FEV_1_	AI Free Choice + FEV_1_
**Fitting**		
Sensitivity–Specificity–MCC	1.00	1.00
Coverage	1.00	1.00
**Prediction**		
Sensitivity	0.83	1.00
Specificity	1.00	0.92
MCC	0.83	0.92
Coverage	0.85	0.96
PPV (rep/our)	1.00/1.00	0.92/0.86
NPV (rep/our)	0.87/0.93	1.00/1.00
Accuracy (rep/our)	0.92/0.95	0.96/0.95

PPV: NPV and accuracy were calculated based on either (a) the mortality observed in our COPD cohort (32%) [our] or (b) the mortality rate previously reported [rep] in the Spanish COPD population (47%) [11], which aligns with mortality predictions derived from the 3CIA initiative detailed in Appendix C [6,12]. Abbreviations: MCC: Matthews correlation coefficient; PPV: positive predicted value; NPV: negative predicted value; rep: reported mortality; our: our own mortality; AI: artificial intelligence.

## Data Availability

All data supporting the findings of this study are available within the article and its Appendix A.

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
