# Peer review of "Metabolomic Signatures Predict Seven-Year Mortality in Clinically Stable COPD Patients"

_ijms, 2025, doi:10.3390/ijms26136373_

Round 1
Reviewer 1 Report
Comments and Suggestions for Authors
This is a well-written and methodologically robust pilot study investigating whether baseline plasma metabolomic profiles can predict long-term mortality in clinically stable COPD patients. The authors combine untargeted and semi-targeted LC-MS approaches with machine learning to develop predictive models, further enhancing them by integrating FEV₁. The biological interpretation of differentially abundant metabolites (DAMs) is rigorous and thoughtfully linked to known COPD pathophysiology, particularly mitochondrial dysfunction, energy metabolism, redox balance, and microbiota-host interactions. Despite the small sample size, the work holds significant promise for translational biomarker research in COPD prognosis. However, the paper merits publication after revisions addressing the following concerns:
- While the study is clearly labeled as a pilot, the small final cohort (n=41) raises concerns about overfitting and generalizability of the predictive models. Although the use of cross-validation and conformal prediction partially mitigates this, the authors should provide more detailed justification or power calculation results in the main text and clarify whether external validation is planned.
- The cohort is limited to non-frequent exacerbators and predominantly older male patients from Spain. This restricts the generalizability of findings. The manuscript should discuss this limitation more explicitly, especially in terms of sex and exacerbator phenotype bias.
- The study does not account for treatment changes, exacerbations, or clinical events during the 7-year follow-up, which could significantly influence outcomes and metabolite levels. Although acknowledged as a limitation, a sensitivity analysis excluding early deaths or stratifying by key treatments (e.g., inhaled corticosteroids) would strengthen conclusions.
- The manuscript is clearly written, with fluent academic language. Occasional grammatical issues are minor and do not hinder comprehension. However, professional language editing may further improve clarity in certain parts.
Author Response
Response to Reviewer #1
This is a well-written and methodologically robust pilot study investigating whether baseline plasma metabolomic profiles can predict long-term mortality in clinically stable COPD patients. The authors combine untargeted and semi-targeted LC-MS approaches with machine learning to develop predictive models, further enhancing them by integrating FEV₁. The biological interpretation of differentially abundant metabolites (DAMs) is rigorous and thoughtfully linked to known COPD pathophysiology, particularly mitochondrial dysfunction, energy metabolism, redox balance, and microbiota-host interactions. Despite the small sample size, the work holds significant promise for translational biomarker research in COPD prognosis.
Response: We thank the reviewer for his/her thoughtful and encouraging evaluation of our work. We are pleased that the manuscript was considered well-written and methodologically robust, and that the reviewer acknowledged the rigorous integration of untargeted and semi-targeted LC-MS analyses, the machine learning approach, the added prognostic value of incorporating FEV1, and the discussion of the pathophysiological implications of our results. We sincerely appreciate the reviewer’s recognition of the promise of our study.
- While the study is clearly labeled as a pilot, the small final cohort (n=41) raises concerns about overfitting and generalizability of the predictive models. Although the use of cross-validation and conformal prediction partially mitigates this, the authors should provide more detailed justification or power calculation results in the main text and clarify whether external validation is planned.
Response: We appreciate the reviewer’s concern regarding the small sample size. As noted in the manuscript, this is a pilot study specifically designed to explore potential metabolic predictors of long-term mortality in clinically stable COPD patients, with a non-frequent exacerbator phenotype. The sample size was calculated based on effect sizes from a previous proteomics study using the GRANMO software, as detailed in Section 4.4.1.
To mitigate the risk of overfitting, we employed robust statistical techniques (limma with SVA adjustment) and a conformal prediction variant of Random Forest with 5-fold cross-validation. Furthermore, the consistent biological plausibility of the differentially abundant metabolites (DAMs) and the strong model performance provide additional support for the validity of the findings.
We fully agree that external validation is essential. Accordingly, we have now included a statement in the Discussion (section 3.6) outlining our plans to replicate these results in an independent, larger cohort using a targeted metabolomics approach.
- The cohort is limited to non-frequent exacerbators and predominantly older male patients from Spain. This restricts the generalizability of findings. The manuscript should discuss this limitation more explicitly, especially in terms of sex and exacerbator phenotype bias.
Response: We thank the reviewer for this observation. While we agree that the predominance of older male non-frequent exacerbators may limit generalizability of our findings to other COPD subgroups, we believe this design choice also represents a methodological strength. Given the clinical and biological heterogeneity of COPD, we deliberately focused on a more homogeneous and clinically stable phenotype to reduce variability and enhance signal detection. Non-frequent exacerbators constitute a substantial proportion of the global COPD population — indeed, this phenotype accounts for the vast majority of COPD patients—and are particularly well suited for long-term prognostic biomarker research due to their relative clinical stability.
On the contrary, including patients with frequent exacerbations would have introduced a bias toward high early mortality, which is clinically predictable. The strength of this study lies precisely in its ability to identify patients whose current clinical condition does not suggest a poor outcome, with the aim of offering them more targeted care.
Importantly, our conceptual framework aligns with the broader shift toward personalized medicine in COPD. Rather than aiming for a single, generic model applicable to all patients—which may obscure biologically meaningful signals—we advocate for the development of phenotype-specific models tailored to well-defined clinical subgroups. In this context, our findings provide novel evidence that even among seemingly lower-risk non-exacerbator patients, early metabolomic alterations can predict long-term mortality with high accuracy. This supports the notion that personalized risk stratification is both feasible and clinically valuable, even in milder disease phenotypes.
The male predominance in our cohort reflects the demographic characteristics of COPD patients in Spain during the recruitment period. Nevertheless, we ensured the inclusion of women in the study, and importantly, the final sex distribution did not differ significantly between survivors and deceased patients. We have now explicitly addressed these considerations in the revised Discussion (Section 3.6) to provide a more nuanced interpretation of the study’s strengths and limitations, particularly in relation to sex and phenotype-specific generalizability.
- The study does not account for treatment changes, exacerbations, or clinical events during the 7-year follow-up, which could significantly influence outcomes and metabolite levels. Although acknowledged as a limitation, a sensitivity analysis excluding early deaths or stratifying by key treatments (e.g., inhaled corticosteroids) would strengthen conclusions.
Response: We fully agree with the reviewer that longitudinal clinical events—such as treatment modifications, exacerbations, or intercurrent illnesses during follow-up—may influence long-term mortality outcomes. However, we would like to emphasize that all metabolomic samples were collected at baseline, prior to the onset of the 7-year follow-up period. Therefore, post-baseline events could not have influenced plasma metabolite composition at the time of sampling and thus cannot directly confound the metabolomic profiles used for prediction. While these variables may have acted as mediators along the pathway to mortality, they cannot impact the results a priori (See also our response to comment #2).
However, in response to the reviewer’s valuable suggestion, we performed a post hoc analysis excluding patients who died within the first 2 years of follow-up (n = 4). This yielded a subcohort of 38 patients (28 survivors, 10 deceased between years 2 and 7). Differential analysis in this subset showed that the majority of DAMs remained consistent in direction and similar in magnitude of change, with p-values <0.1 despite the reduced statistical power for key features (see new Supplementary Table S2).
We also acknowledge the reviewer’s suggestion to stratify by treatment class, particularly inhaled corticosteroids (ICS). Unfortunately, detailed longitudinal treatment data were not consistently available across all centers. However, current guidelines do not recommend ICS use in non-frequent exacerbators. This indication applies only to frequent exacerbators and to patients with blood eosinophil levels >300 cells/µL (19.5% in our series). Therefore, it is unlikely that ICS use was either frequent or a major determinant of our results. Nonetheless, we have now made this limitation more explicit in the revised Discussion and fully agree that future prospective studies with repeated sampling and structured follow-up will be critical to validate and extend these findings.
- The manuscript is clearly written, with fluent academic language. Occasional grammatical issues are minor and do not hinder comprehension. However, professional language editing may further improve clarity in certain parts.
Response: We are grateful to the reviewer for his/her positive assessment of the manuscript's overall clarity and fluency. In response to the suggestion, we have revised the text with the assistance of a native English-speaking collaborator to further improve linguistic precision and flow. We trust that these edits have enhanced the manuscript’s readability without altering its scientific content.
Reviewer 2 Report
Comments and Suggestions for Authors
In this manuscript, the authors present targeted and semi-targeted LC-MS approaches to predict survival outcome of a random sample of COPD patients. While the work is interesting, several major points need to be addressed:
- What is the significance of selecting "random" sample of 54 patients in the initial set? How was it ensured that the chosen set is random in the absence of any apparent randomized data set? I am confused by this logic and the conclusion inferred.
- What statistical tests were performed to calculate p-values of significance? Again, in the apparent absence of positive and/or negative controls, it is difficult to infer true statistical significance of the results presented.
- How would the results be biased if the selected patient sample were chosen based on certain criteria, such as age, gender etc.?
- Would the significant and non-significant metabolites shown in Fig 1 cluster clearly or would there be overlap? If yes, why?
- How were the p-values in Table 2 calculated? Can the authors calculate ROC curve by determining true positive and false negative values?
- I am confused how the network analysis in Fig 3 was done. How were the direction of change and magnitude of difference were calculated?
- Table 4 needs more explanation regarding how the "conventional" and "AI free choice" parameters were calculated from the input data.
Author Response
Response to Reviewer #2
- What is the significance of selecting "random" sample of 54 patients in the initial set? How was it ensured that the chosen set is random in the absence of any apparent randomized data set? I am confused by this logic and the conclusion inferred.
Response: We thank the reviewer for highlighting this important point. The term "random" was used to indicate that the 54 patients included in the study were selected without prior stratification or outcome-based filtering from the pool of clinically stable, non-frequent exacerbators within the larger BIOMEPOC cohort. Although the process did not involve a formal computer-generated randomization algorithm, the initial selection—conducted at the start of the study—was performed without applying any clinical, demographic, or survival-related criteria. This approach helped minimize selection bias and preserved the exploratory nature of this pilot study. It is also important to note that all COPD patients were smokers or ex-smokers, as we applied the C-COPD (classical/smoker-linked COPD) definition.
The rationale for limiting the sample size was both financial and logistical, as untargeted LC-MS metabolomics is a resource-intensive and costly approach. Accordingly, this study was designed as a pilot investigation aimed at identifying potential prognostic signatures could justify validation in larger, independently sampled cohorts.
To avoid misunderstandings, we have revised the Methods section (Section 4.1) to further clarify that patients selection was non-stratified and pragmatically random, and to provide a more detailed explanation of the rationale behind this design choice.
- What statistical tests were performed to calculate p-values of significance? Again, in the apparent absence of positive and/or negative controls, it is difficult to infer true statistical significance of the results presented.
Response: We thank the reviewer for raising this methodological point. As clarified in the Methods section (Section 4.4.4), we performed univariate differential analysis using moderated t-tests with empirical Bayes variance shrinkage, implemented via the limma package in R, to compare metabolite abundances of metabolites between patients who died and those who survived after the 7-year follow-up period. To control for potential confounders, the models were adjusted for age, post-bronchodilator FEV₁, and surrogate variables (SVs), thus accounting for latent sources of variability in the metabolomic data. We have revised this section to improve its clarity.
We acknowledge that positive and negative controls are often lacking in metabolomics studies involving human subjects, particularly when investigating clinical outcomes such as mortality. Nevertheless, rigorous internal normalization, exclusion of outliers, robust preprocessing, and downstream modeling can substantially reduce the likelihood spurious significance.
- How would the results be biased if the selected patient sample were chosen based on certain criteria, such as age, gender etc.?
Response: We appreciate the reviewer’s thoughtful question. Indeed, selecting patients based on predefined criteria such as age, sex, or disease severity could introduce selection bias and confounding, potentially distorting associations between metabolite profiles and mortality.
To mitigate this risk, the sample was selected without any stratification by clinical or demographic variables at the beginning of the study, prior to the occurrence of any outcomes. In the statistical analysis, we further addressed potential confounding through covariate adjustment—particularly for age and post-bronchodilator FEV₁—using surrogate variable analysis (SVA) in the differential metabolomic analysis, as well as through feature selection and validation procedures in our predictive modeling pipeline.
While we acknowledge that residual confounding is an inherent limitation of observational studies, these methodological safeguards help reduce the likelihood of spurious associations. We have now explicitly incorporated this conceptual limitation and the rationale for our mitigation strategies in the revised Discussion (Section 3.6).
4 Would the significant and non-significant metabolites shown in Fig 1 cluster clearly, or would there be overlap? If yes, why?
Response: Some degree of overlap is expected in both unsupervised and supervised projection methods (such as PCA or PLS-DA), due to the biological heterogeneity and technical noise inherent to high-dimensional metabolomics data. For this reason, we did not rely on linear projection models for classification. Instead, we employed Random Forest with embedded feature selection—a method that is more robust to such variability. It is important to note that complete separation of groups would imply a perfect classification model (i.e., 100% accuracy, sensitivity, specificity, and MCC = 1), which is rarely attainable in biological datasets. In this context, the performance metrics we report are high but realistically reflect the complexity of the phenotype.
To further explore the clustering structure of the most relevant features, we generated a heatmap based on the 12 statistically significant differentially abundant metabolites (DAMs) listed in Table 2). These included: butyryl-L-carnitine, levodopa, hydroxyproline, indoxyl sulfate, phenylacetylglutamine, monomethylglutarate, creatine, pyruvic acid, L-cystine, 1-methylnicotinamide, 4-aminobenzoate, and citric acid. Hierarchical clustering was performed on autoscaled, normalized data using Minkowski distance and complete linkage. As expected, some overlap is observed between survival groups, reflecting biological variability. Nonetheless, the fact that models built on these DAMs outperform those using arbitrarily selected features (Tables 3 and 4) supports their discriminative and biological relevance, even if they are not fully separable in unsupervised visualizations.
See the attached Heatmap (in the pdf file) of the 12 differentially abundant metabolites (DAMs) between deceased and surviving COPD patients after 7-year follow-up. Data were autoscaled and clustered using Minkowski distance and complete linkage.
- How were the p-values in Table 2 calculated? Can the authors calculate ROC curve by determining true positive and false negative values?
Response: We thank the reviewer for this question. As noted in our response to Comment #2, the p-values reported in Table 2 were calculated using moderated t-tests with empirical Bayes adjustment, implemented via the limma package in R. These tests were performed on log₂-transformed, normalized metabolite intensities and adjusted for age, post-bronchodilator FEV₁, and surrogate variables (SVs) to account for potential confounding and latent sources of variability.
Regarding ROC curves, we initially did not include them because our primary classification approach is based on Random Forest with conformal prediction, which generates discrete prediction sets rather than continuous probability scores. As such, traditional ROC curve analysis is not directly applicable to this framework. Instead, we reported specificity, sensitivity, Matthews correlation coefficient (MCC), and predictive values, which are more appropriate for models whose performance is not dependent on decision thresholds.
Nevertheless, in response to the reviewer’s request, we generated ROC curves based on the global performance of the models (Tables 3 and 4), providing an approximation of AUROC values. The resulting curves and corresponding metrics have been included in Supplementary Figure S1, and this addition is now described in the revised Results (Section 2.4).
- I am confused how the network analysis in Fig 3 was done: How were the direction of change and magnitude of difference were calculated?
Response: We appreciate the reviewer’s interest in the network visualization presented in Figure 3. This figure was designed to illustrate the functional relationships among DAMs between deceased and surviving COPD patients after the 7-year follow-up, as listed in Table 2.
The direction of change was determined based on the log₂ fold change (logFC) in metabolite abundance between groups. In the network, this is conveyed through node color intensity: darker tones indicate higher levels in deceased patients, while lighter tones indicate lower levels.
The magnitude of the difference was calculated as the absolute percentage change between groups and is represented by node size: larger nodes correspond to metabolites with greater percentage differences.
Edges in the network denote shared biological functions or pathway involvement, based on curated annotations from the SMPDB, KEGG, and HMDB databases. This mapping was performed manually through expert curation.
The final network was visualized using Cytoscape software (v3.10.2, www.cytoscape.org). These methodological details are now included in the Methods (Section 4.5.5) and summarized in the revised Figure 3 legend for clarity.
- Table 4 needs more explanation regarding how the "conventional" and "AI free choice" parameters were calculated from the input data.
Response: As explained in the Methods (Sections 4.5.2–4.5.4), the "conventional" model used the top 10 DAMs identified through univariate analysis, whereas the "AI free choice" model allowed the algorithm to select the most informative features via embedded feature selection, specifically K-best feature selection. Table 4 presents a comparison of these approaches with integration of post-bronchodilator FEV₁.
As further detailed in Section 4.5.4 of the Methods. model predictions were compared with actual survival outcomes using a confusion matrix, from which the following performance metrics were calculated: sensitivity (SE), specificity (SP), positive predictive value (PPV), negative predictive value (NPV), overall accuracy (Acc), and the Matthews correlation coefficient (MCC, also referred to as the phi coefficient, φ or rφ).
We have expanded the explanation at the first mention of Table 3 in the Results (Section 2.4) to improve clarity.

Round 2
Reviewer 2 Report
Comments and Suggestions for Authors
The revised manuscript addresses the pertinent questions raised by this reviewer.